# New Hyaluronic Acid/Polyethylene Oxide-Based Electrospun Nanofibers: Design, Characterization and In Vitro Biological Evaluation

**DOI:** 10.3390/polym13081291

**Published:** 2021-04-15

**Authors:** Oana Maria Ionescu, Arn Mignon, Andreea Teodora Iacob, Natalia Simionescu, Luminita Georgeta Confederat, Cristina Tuchilus, Lenuța Profire

**Affiliations:** 1Department of Pharmaceutical Chemistry, Faculty of Pharmacy, Grigore T. Popa University of Medicine and Pharmacy of Iași, 16 University Street, 700028 Iasi, Romania; oana-maria.dc.ionescu@d.umfiasi.ro (O.M.I.); andreea.panzariu@umfiasi.ro (A.T.I.); 2Smart Polymeric Biomaterials, Surface and Interface Engineered Materials, Campus Group T, KU Leuven, Andreas Vesaliusstraat 13, 3000 Leuven, Belgium; 3Centre of Advanced Research in Bionanoconjugates and Biopolymers, Petru Poni Institute of Macromolecular Chemistry, 41A Grigore Ghica Voda Alley, 700487 Iasi, Romania; natalia.simionescu@icmpp.ro; 4Emergency Clinical Hospital “Prof. Dr. Nicolae Oblu”, 2 Ateneului Street, 700309 Iasi, Romania; 5Department of Microbiology, Grigore T. Popa University of Medicine and Pharmacy of Iasi, 700115 Iasi, Romania; georgeta-luminita.confederat@umfiasi.ro (L.G.C.); cristina.tuchilus@umfiasi.ro (C.T.)

**Keywords:** hyaluronic acid, polyethylene oxide, electrospinning, nanofibers, wound dressings

## Abstract

Natural compounds have been used as wound-healing promoters and are also present in today’s clinical proceedings. In this research, different natural active components such as propolis, Manuka honey, insulin, L-arginine, and *Calendula officinalis* infusion were included into hyaluronic acid/poly(ethylene)oxide-based electrospun nanofiber membranes to design innovative wound-dressing biomaterials. Morphology and average fiber diameter were analyzed by scanning electron microscopy. Chemical composition was proved by Fourier transform infrared spectroscopy, which indicated successful incorporation of the active components. The nanofiber membranes with propolis and *Calendula officinalis* showed best antioxidant activity, cytocompatibility, and antimicrobial properties against pathogen strains *Staphylococcus aureus, Escherichia coli,* and *Pseudomonas aeruginosa* and had an average diameter of 217 ± 19 nm with smooth surface aspect. Water vapor transmission rate was in agreement with the range suitable for preventing infections or wound dehydration (~5000 g/m^2^ 24 h). Therefore, the developed hyaluronic acid/poly(ethylene)oxide nanofibers with additional natural components showed favorable features for clinical use as wound dressings.

## 1. Introduction

The acute and chronic wound care management represents a major health problem in the entire world, and biomedical research is constantly seeking new options with respect to promoting the healing process and to reducing the therapy cost. In 2021, the wound therapy cost is estimated to exceed 20 billion dollars [1]. Therefore, a cost-effective therapy is needed to aid the patient overcome this health issue in a shorter timeframe to prevent further complications that can debilitate someone’s social, economic, and mental wellbeing [1].

The dynamic process of wound healing needs the support of a proper dressing that can mimic and promote the natural process of healing [2]. While it is important to keep a moist environment that can aid healing, the designing of a dressing that would meet the demanding qualities for maintaining cell growth whilst ensuring their differentiation, still represents an important aspect of biomaterial engineering. Different dressing biomaterials such as films, hydrogels, sponges, and electrospun membranes are studied for their properties as a support in wound healing [3]. These materials are characterized for their surface properties and morphology that are highlighted using scanning electronic microscopy, transmission electronic microscopy, or fractal model for porous membranes [4,5,6].

Nanofibers (NFs) are relatively new porous systems currently investigated [7,8] for their multiple purposes. Based on their unique properties (large surface and high porosity), NFs act as excellent extracellular matrices [9] which enhance the tissue formation. Mass transport through porous nanofibers is also an important characteristic and using a mathematical model based on fractal character of micropores, the imbibition height and the imbibition mass could be calculated [5]. Designing NFs delivers a large range of possibilities in the matter of selecting the polymers and active components (ACs) [10,11].

In the case of wound healing, excessive production of reactive oxygen species may lead to slow regeneration of the tissue. Vitamins, catechol-based products, and glutathione proteins have been studied for their antioxidant activity and presented benefits when added to wound dressings [12,13]. An antimicrobial effect is a desirable property of a wound-dressing material since it can diminish the inflammatory response to local infection and help with the ever-growing problem of antibiotic resistance [14].

The rationale of the wound dressings developed in this study starts from having a polymeric matrix, based on hyaluronic acid (HA), that not only protects the wound from external damaging agents yet adds value through biological effects, with HA promoting the cellular proliferation. The active components chosen to be incorporated were taken into consideration for their benefits in speeding the healing process, because of the multitude of risks associated with wounds. Therefore, re-epithelization agents (Manuka honey and insulin), antimicrobial agents (propolis, Manuka honey), an angiogenetic compound (l-arginine), and an antioxidant/re-epithelization agent (*Calendula officinalis* infusion) were selected for this study.

Hyaluronic acid (HA) is a glycosaminoglycan found in high concentrations in connective tissue and skin. Chemically, it is a linear polysaccharide made of repeating units of the following disaccharides: 1-4-d-glucuronic acid and 1-2-*N*-acetyl-d-glucosamine [15]. Shortly after the wound has occurred, HA may act as a temporary supporting structure, as a result of its high molecular weight and viscoelasticity [16]. Therefore, it permits nutrients and cells (keratinocytes) to diffuse easily. In the first phase of wound healing (haemostasis), HA combines with fibrin and forms a compound that accelerates cell’s migration. It can also stimulate cutaneous adnexa (follicles, apocrine or sebaceous glands). Unfortunately, HA has low electrospinning properties by itself, and various successful attempts have been made using additional compounds to convert it into NFs [6,17,18]. In water, the HA chains form an expanded coil and low concentration solutions deliver polymer chain entanglements. The higher the concentration conducts, the higher the viscosity, which is not beneficial for the electrospinning process. To provide lower viscosity and successful electrospinning of HA solutions, sodium chloride and PEO was used as viscosity modifiers. This blended HA solution is less viscous, has a pH level above 4, and provides successful NFs [18].

Polyethylene oxide (PEO) is a synthetic, biocompatible, biodegradable, hydrophilic polymer and is easily soluble in water. It is often used in the fabrication of NFs through the electrospinning technique and mostly plays the role of a copolymer in the formulation. 

Although different HA-PEO nanofibers have been previously prepared [6,19] into blend or core-shell structures, none have yet incorporated Manuka honey, propolis, *Calendula officinalis* infusion, and l-arginine to our knowledge.

The main component of Manuka honey is methylglyoxal (MGO). Its immunomodulatory action during healing is associated with the release of cytokines from leucocytes, thus regulating proliferation of fibroblasts. It is in fact a natural derivative with re-epithelization and antibacterial effects at the wound site. By reducing inflammation (via decreasing the reactive oxygen species scavenging activity) and absorbing wound exudate through its osmotic mechanism [20], it prevents maceration while promoting angiogenesis and granulation tissue production. The final effects are accelerating wound contraction and epithelialization [21,22].

Propolis is a resinous product made by bees. Through its complex chemical composition, it displays numerous biological effects. Local flora, climate condition, weather, and other components such as wax or pollen may influence the chemical composition. Of high interest from a biological point of view are the flavonoids, phenolic acids, and their ester derivatives. The literature is abundant of examples of applications for propolis based on its antioxidant, anti-inflammatory, and antimicrobial properties. Propolis was previously incorporated into NFs for the treatment of leg ulcers [23,24].

L-arginineis a basic α-amino acid that plays several key roles in the physiology of the cell. Its alkaline guanidine group (carbon-nitrogen double bond) has reductive effects that can enhance antioxidant activity in wound-dressing materials [13]. It may also be considered as a nitric oxide precursor [25] that is responsible for local responses at the wound site [26] and is involved in granulation tissue formation and also epithelialization [27]. It was noticed that low concentrations of L-arginine are present at the wound site because of the arginase enzyme which is responsible for the hydrolysis of L-arginine to degradation compounds (urea and L-ornithine) [28].

Insulinis a hormone which acts as a growth factor and promotes healing at the wound site [29]. When applied to a wound, insulin promotes the production of granulation tissue, speeds epithelialization, and wound contraction [30].

*Calendula officinalis* is often used in therapy as an infusion, tincture, or ointment when treating inflammation of the skin, burns, insect bites, and leg ulcers. It displays curative properties due to its flavonoids and saponins [31,32,33].

Although propolis has already been successfully incorporated into polyurethane-HA [34] and L-arginine into polyvinylalcohol-HA [35] scaffolds, no combination of them with other ACs, such as insulin, Manuka honey, and *Calendula officinalis* has been yet reported. Many benefits may derive from combination of ACs that can enhance the biological properties of HA with respect to regenerative medicine. In the emerging concept of biomimetism, which is adapting principles from nature into science and medicine, we report the preparation, physicochemical characterization and in vitrobiological evaluation of HA-PEO-NFs, with biologically ACs, such as Manuka honey, propolis, *Calendula officinalis* infusion, insulin, and L-arginine, blended into the polymer matrix in different combinations, as possible candidate materials for wound applications.

## 2. Materials and Methods

### 2.1. Materials

Hyaluronic acid (HA) sodium salt from *Streptococcus equi,* with a molecular weight of 1.5 × 10^6^ Da (which is classified as high molecular weight) and poly(ethylene oxide) (PEO), with a molecular weight of 200 kDa, L-arginine∙HCl, DPPH (2,2-Diphenyl-1-picrylhydrazyl), α-bromo-naphtalene (97%), ABTS (2,2’-azino-bis(3-ethyl-benzthiazoline-6-sulfonic acid), disodium hydrogen phosphate, sodium chloride, and sodium dihydrogen phosphate, were purchased from Sigma-Aldrich (Merck Group, Schnelldorf, Germany). Insulin (Novo Rapid Penfill 100 IU/mL, (Novo Nordisk, Bagsvaerd, Dennmark), Manuka honey 550+ (Manuka HealthLtd., Te Awamutu, New Zealand), *Calendula officinalis flos* (Fares, Orăştie, Romania), and propolis (Apiland, Baia Mare, Romania) were incorporated into the nanofibers. Gram-negative (*Escherichia coli* ATCC 25922, *Pseudomonas aeruginosa* ATCC 27853), and Gram-positive (*Staphylococcus aureus* ATCC 25923) bacterial strains as well as pathogenic yeasts (*Candida albicans* ATCC 10231) were provided by the Department of Microbiology of Grigore T. Popa University of Medicine and Pharmacy of Iasi, Romania. All materials were used without further purification. A sterile, isotonic, and aqueous solution of sodium chloride (0.9%) (B. Braun^®^, Melsungen, Germany) was purchased. Alpha-MEM medium (Lonza, Basel, Switzerland), fetal bovine serum (FBS, Gibco, Thermo Fisher Scientific, Waltham, MA, USA), and 1% penicillin-streptomycin-amphotericin B mixture (10 K/10 K/25 μg, Lonza, Basel, Switzerland) were also used.

### 2.2. Preparation of HA-PEO-ACs Solutions

Firstly, the HA was dissolved in normal saline solution, and then PEO was added, and a homogeneous polymeric solution was obtained after 24 h of stirring (EcoStir, DLAB Scientific Inc., Ontario, CA, USA). The molecular weight of PEO is a key factor in obtaining HA-PEO-NFs. Previous attempts which have included PEO 1000 kDa proved unsuccessful in obtaining NFs. When PEO 200 kDa was used, the electrospinnability of the polymeric solutions was optimal, and successful HA-PEO-NFs were obtained. The total polymer concentration was kept at 12.5% (wt/wt). Afterwards, different ACs, such as Manuka honey (M), propolis (P), insulin (I), and L-arginine (L), were added. The electrospinning solutions were blended on a tabletop shaker (MaxQ HP Tabletop Shaker, ThermoFisher Scientific, Waltham, MA, USA) until a homogenous composition was reached. Afterwards, the solutions were left to rest for 1 h for degassing at room temperature. Detailed data on the electrospinning solutions are listed in Table 1. To prepare the HA_PEO@PC solution, slight modifications were applied to the protocol: an infusion of *Calendula officinalis flos* (C) was firstly prepared according to the EMEA monograph from 1.5 g of *flos* and 150 mL water (70 °C). Then, the solution was filtered and kept at room temperature and used in the first 24 h as a solvent for HA, PEO, and propolis.

### 2.3. The Viscosity Measurements of HA-PEO-ACs Solutions

The rheological properties of the HA-PEO-ACs solutions were studied using the Ostwald model to show the pseudoplastic behavior and the Carreau–Yassuda model to determine the zero-shear viscosity. A rheometer from Anton Paar (Physica MCR, Graz, Austria) with a 50 mm upper plate set in a parallel-plate system was used. The flow curves were carried out with shear-rates ranging from 0.01 to 500 s^−1^. Before each determination, the solutions were firstly equilibrated at 25 °C. The apparent viscosity of the polymeric solutions was determined at 100 s^−1^.

### 2.4. Preparation of HA-PEO-NFs

Referring to HA-PEO blending solution, HA is a negatively charged polymer, whose carboxyl groups are ionized in carboxylate ions in the normal saline, which serves as an aqueous solvent and endows the HA electrical properties. Under the electrostatic force, the HA molecules move along the opposite direction of the electric field. Electrospinning was carried out using the Nanospinner Inovenso (Inovenso Ltd., Istanbul, Turkey) equipment, which is composed of an infusion pump, a generator, and a negative electrode under the form of a plate collector. Electrospinning parameters were kept constant during the experiments, with a 13–15 cm distance from the needle tip to the collector, a flow rate of 0.5–0.7 mL/h, and a voltage below 20 kV.

### 2.5. Characterization of the HA-PEO-NFs

#### 2.5.1. Surface Morphology and Fiber Diameter

Fiber morphology was carried out using scanning electron microscopy (SEM). The micrographs were performed with a Phenom Desktop SEM-FEI (Hillsboro, OR, USA). Dry samples were set to a sample holder with a double-sided carbon tape, and then gold-sputter-coated (20 mA, 60 s, vacuum) by an automatic Au sputter coater EmiTech K550X (EmiTech Ltd., Ashford, UK) with a RV3 two-stage rotary vane pump. Fiber diameter and distribution were analyzed using the phenom fiber metric software (Phenom Pro, Eindhoven, the Netherlands). At least three images per sample and 150 fibers per image were measured. Data are shown as average diameter ± standard deviation (SD).

#### 2.5.2. Fourier Transform Infrared Spectroscopy (FTIR)

The infrared spectra of the developed HA-PEO-NFs were recorded using the IR ABB MB300 spectrophotometer (ABB, Zurich, Switzerland). All spectra were collected in the wave number range 4000–600 cm^−1^ at a resolution of 4cm^−1^.

#### 2.5.3. Water-Vapor Transmission Rate (WVTR)

In order to measure the HA-PEO-NFs moisture permeability, the evaluation of WVTR was carried out according to the standard water method assayed by the American Society for Testing and Materials (ASTM 95-96) and European Pharmacopoeia, with slightly modifications [36]. All samples were dried before the test, then cut (3 cm diameter samples), and set on the mouth of glass vials (of 2 cm diameter) which contained 10 mL of distilled water each. The NFs were secured with paraffin tape, alongside the vials’ edges. Afterward, the vials were placed in an incubator (Roth M, Karlsruhe, Germany) and kept at constant temperature and humidity (37 °C, to mimic physiological temperature). The rate of change of mass in the water from the vials over which the NFs were mounted was recorded. The weight change was measured, and the WVTR was calculated according to the following equation:WVTR (g/m^2^·24 h) = (W_i_−W_f_)/A × t(1)
where W_i_ and W_f_ represent the initial and final weight of bottles containing water and A is the exposure area. The experiment was performed in triplicate, and the results are expressed as mean ±SD.

#### 2.5.4. In Vitro Antioxidant Assays

The radical scavenging activity of HA-PEO-NFs toward DPPH and ABTS^•+^ was measured. Then, the total antioxidant capacity and also ferric-reducing antioxidant power were determined. Samples of 60 mg, corresponding to the HA-PEO-NFs, were immersed in 5 mL ethanol, and a set volume of the ethanolic extract was used for experiments. A GBC Cintral UV–Vis spectrophotometer (GBC Scientific Equipment, Braeside, Victoria, Australia) was used to perform measurements. All experiments were performed in triplicate, and the results are expressed as mean ± SD.

DPPH Radical Scavenging Assay

The DPPH assay is one of the most used methods to determine the radical scavenging effects. The violet color of DPPH in ethanolic solution turns into a yellow in the presence of a proton-donating agent [37], indicating the reduction of the DPPH (Figure 1).

In brief, a volume of 0.25 mL of the ethanolic extract was mixed with 1 mL of 100 μM DPPH ethanolic solution. The obtained solution was kept in the dark at room temperature. After 60 min, the absorbance was measured at 517 nm, and the inhibition (%) was calculated according to the following equation:Inhibition (Scavenging activity) % = [(A_DPPH_− A_s_)/A_DPPH_] × 100(2)
where A_s_ represents the absorbance of the HA-PEO-NFs sample and A_DPPH_ is the absorbance of the DPPH ethanolic solution.

ABTS^•+^ Radical Scavenging Assay

The ABTS (7 mM) aqueous solution was prepared with ammonium persulfate (2.45 mM) and was kept in the dark for 16 h in order to initiate the formation of the ABTS^•+^. The literature protocols indicate that the solution be diluted with ethanol to obtain an absorbance of 0.700 ± 0.02 at 734 nm [38]. The blue chromophore ABTS^•+^is reduced in the presence of hydrogen donating agents (Figure 2). The result is the decrease of the absorbance [39].

A volume of 1 mL ABTS solution was added to 0.25 mL of HA-PEO-NFs ethanolic extract. After 6 min, the absorbance of the samples was measured at 734 nm against a blank (ethanol was used instead of the set volume of HA-PEO-NFs ethanolic extract). The inhibition (%) representing the ABTS^•+^radical-inhibiting capacity was calculated according to the following equation [38,39]:Inhibition (Scavenging activity) % = [(A_ABTS_ − A_s_)/A_ABTS_] × 100(3)
where A_ABTS_ is absorbance of the ABTS^•+^ solution; A_s_ is absorbance of the HA-PEO-NFs sample.

Phosphomolybdenum-Reducing Antioxidant Power (PRAP) Assay

The total antioxidant capacity of HA-PEO-NFs was assayed using the modified phosphomolibdenum method, as was previously described in literature [40]. This spectrophotometric determination is based on the reducing of Mo(VI) to Mo(V) in the presence of electron donating agents in acidic medium [41]. The resulting phosphomolibdenum complex is of green color. Briefly, a volume of 2 mL reagent solution (consisting of 0.6 M sulfuric acid, 4 mM ammonium molibdate, and 28 mM disodium hydrogen phosphate) was mixed with 0.2 mL of HA-PEO-NFs ethanolic extract. All samples were then incubated at 95 °C for 90 min. After cooling (reaching room temperature), the absorbance was measured at 695 nm against the blank (where ethanol was used instead of the set volume of HA-PEO-NFs extract). The increase of optical density implies superior antioxidant effects.

Ferric-Reducing Antioxidant Power (FRAP) Assay

The ferric-reducing antioxidant power method is based on the reduction of ferricyanide into ferrocyanide (blue), in the presence of electron donating agents [31]. To 0.5 mL HA-PEO-NFs extract, 0.5 mL of 0.2 M phosphate buffer (pH = 6.6) were added. To initiate the reaction, 0.5 mL of potassium ferricyanide 1% (wt/*V*) were added, and the samples were incubated at 50 °C for 20 min. To complete the reaction, 0.5 mL of trichloroacetic acid 10% (wt/*V*) were added. In addition, 1 mL of double distilled water and 0.2 mL of ferric chloride 0.1% (wt/*V*) were added to 1 mL of the previously resulting solution. The samples were left to rest for 10 min at room temperature, and their absorbance was measured at 700 nm against a blank (consisting of ethanol instead of the set volume of HA-PEO-NFs ethanolic extract). As the case of the PRAP assay, the increase of optical density implies superior antioxidant effects.

#### 2.5.5. In Vitro Cytotoxicity (MTS) Assay

The cytotoxicity degree of HA-PEO-NFs was determined by MTS assay, by using the CellTiter 96^®^aqueous one-solution cell proliferation assay (Promega, Madison, WI, USA). The method was according to the manufacturer’s set of instructions and a procedure adapted from ISO 10993-5 [42]. Normal dermal fibroblast cells were grown in alpha-MEM medium supplemented with 10% fetal bovine serum and 1% penicillin-streptomycin-amphotericin B mixture and seeded at a density of 0.5 × 10^5^ cells/mL into 96-well plates treated with tissue culture and were allowed to adhere for 24 h. The cells were afterwards incubated for 72 h with 100 μL of different concentrations of HA-PEO-NFs ethanolic extract (12.5, 25, 50, and 100 µg/mL) or with fresh complete medium (which also served as control). A volume of 20 µL of the MTS reagent (tetrazolium inner salt) was added to the samples, and after 3 h, the absorbance was read at 490 nm on a FLUOstar^®^ Omega microplate reader (from BMG LABTECH, Ortenberg, Germany). The viability of the cells was expressed as a percentage (%) of the control cells’ viability. The experiment was performed in triplicate, and the results are expressed as mean ± SD.

#### 2.5.6. Antimicrobial Assay

The antimicrobial assay consisted of determination of antibacterial and antifungal activity, by measuring the diameter of the inhibition area, according to the agar disc diffusion method [43]. The Gram-negative (*Escherichia coli* ATCC 25922, *Pseudomonas aeruginosa* ATCC 27853) and Gram-positive (*Staphylococcus aureus* ATCC 25923) bacteria were used as bacterial strains and *Candida albicans* ATCC 10231as pathogenic yeast. The discs containing Ciprofloxacin (5 µg/disc) and Voriconazole (1 µg/disc) were used as controls (Whatman plc, Buckinghamshire, UK) for bacterial and yeast strains, respectively. Suspensions of the bacterial and fungal strains (10^6^ CFU/mL in NaCl 0.9%) were inoculated onto the surface of the culture medium spread at a volume of 25 mL/sterile Petri plate (Mueller–Hinton for bacterial strains and Sabouraud for fungal strain, Merck, Darmstadt, Germany). A volume of 200 µL of HA-PEO-NFs aqueous extract was added into sterile stainless-steel cylinders of 5 mm internal diameter and 10 mm height and placed on the agar surface. The plates were incubated at 37 °C for 24 h (antibacterial activity assessment) and at 35 °C for 48 h (antifungal activity assessment), after which the diameters of the inhibition area (mm, including the disc size) were measured.

#### 2.5.7. Data Analysis

All the experiments were performed in triplicate. The statistical significance was calculated with a t-Test and a one-way ANOVA, where a *p*-value ˂ 0.05 was considered statistically significant.

## 3. Results and Discussion

### 3.1. Viscosity Measurements of HA-PEO-ACs Solutions

The rheological characterization of HA-PEO-ACs solutions, based on Carreau–Yassuda model, is shown in Table 2. Adding Manuka honey to the polymeric solution was partly responsible for the higher zero-shear viscosity value (HA_PEO@ML, 1.7 Pa∙s) when compared to the solutions containing insulin, propolis, or *Calendula officinalis* infusion (HA_PEO@IP; 1.5 Pa∙s; HA_PEO@P, 1.2 Pa∙s; HA_PEO@PC, 0.9 Pa∙s). Viscosity values are directly related to the composition of polymeric solution (content, molecular weight of polymer, and solvent type).

### 3.2. Physicochemical Characterization of HA-PEO-NFs

#### 3.2.1. SEM Morphology and Fiber Diameter

The relative humidity (RH) and concentration of HA have been previously reported to affect the electrospinning of PEO-HA solutions [44,45]. At a lower RH, the aqueous solvent evaporates at a faster rate, and the jet is solidified very quickly during the electrospinning process. In addition, it was noticed that high HA content in PEO solutions and low RH leads to the increase of the viscosity of the polymer solution, which decreases the stretching of the jet and forms NFs with a larger diameter [19].Therefore, the assuring constant RH is a key factor to uniform results. The experiments were carried out at 35–40% RH. As depicted in Figure 3, the developed HA-PEO-NFs displayed a bead-free nanostructure, a homogenous network with smooth surfaces, and uniform diameters. The HA_PEO matrix presented an average diameter of 166 ± 11 nm. The adding of Manuka honey has as result increasing of the diameter of the NFs to 201 ± 12 nm (HA_PEO@ML). This is possibly due to an increase in the charge density caused by the highly hydrophilic Manuka honey, which in turn leads to lesser extension of the polymer jet. In the presence of propolis and *Calendula officinalis* infusion, the average diameter of the NFs increased to 217 ± 19 nm (HA_PEO@PC). Generally, the diameter of developed NFs was less than other NFs reported in the literature [43]. This could be due to higher molecular weight of HA which was used in combination with PEO low molecular weight.

#### 3.2.2. Fourier Transform Infrared Spectroscopy (FTIR)

The composition of the HA-PEO-ACs NFs was proved by the IR spectra (Figure 4), and the collected data are in agreement with the literature [46,47]. The characteristic amides I and II groups of HA were attributed to the absorption bands at 1500 to 1670 cm^−1^. Characteristic bands for carbonyl (C=O) groups appear in the HA spectrum at 1638 cm^−1^ (correspondingly to the amide) and at 1420 cm^−1^ (attributed to the –COO stretching). The free OH groups and hydrogen-bonded OH were identified at 3000–3600 cm^−1^. In addition, the pyranose ring vibrations (at 950 and 11,250 cm^−1^) and C–O–C due to the glycosidic bonds (at 1103 cm^−1^) were also identified. The absorption band at 3447 cm^−1^ was attributed to OH and NH stretching while the absorption band at 2925 cm^−1^ to stretching vibrations of C–H. The characteristic bands that indicate the presence of PEO are around at 2885 cm^−1^ corresponding to the methylene stretching and at around 1100 cm^−1^ as a result of combination of methylene group and ether group stretching, also referred to in the literature [48] as the –C–O–C– absorption complex.

Upon the addition of ACs, characteristic bands were noticed for Manuka honey: at 1050 cm^−1^(C–O), 1636 cm^−1^(C=O), according to the literature data [22]. Specific bands for propolis were present at 1610 cm^−1^, 1490 cm^−1^, and 1450 cm^−1^ (due to C=C stretches of aromatic rings) [34]. Absorption bands at 1640 cm^−1^ and at 1530 cm^−1^ are characteristic of two amide groups belonging to insulin [49]. L-arginine presents characteristic –C=N stretching vibration at 1630 cm^−1^ (due to the guanidine group) [23]. *Calendula officinalis* presented characteristic bands at 1050–1030 cm^−1^ in agreement with literature data on its flavonoid content and ether groups [33].

#### 3.2.3. Water-Vapor Transmission Rate (WVTR)

WVTR is a notable factor in the wound-healing process by promoting gas exchange through the material. Wound dehydration may occur with a high WVTR value and lead to formation of a scar. Moreover, low WVTR acts as wound-healing moisture reservoir because of exudates present at the wound bed which may lead to a possible infection [44]. The results of our study show that by adding propolis to the HA_PEO matrix, the WVTR was significantly improved (5119.6 ± 1 g/m^2^ 24 h for HA_PEO@P, 4959.0 ± 1 g/m^2^ 24 h for HA_PEO@IP and 5122.6 ± 2 g/m^2^ 24 h for HA_PEO@PC) when compared with HA_PEO, unlike HA_PEO@ML which presented similar results with HA_PEO (4636.33 ± 4 g/m^2^ 24 h and 4646.0 ± 5 g/m^2^ 24 h, respectively) (Figure 5).

#### 3.2.4. In Vitro Antioxidant Assays

Since the reactive oxygen species (ROS) are often produced at the wound site, an antioxidant effect is a desirable property for a wound dressing that promotes the healing process [4,50]. The analysis of the data presented in Figure 6 revealed that the best antioxidant effect was obtained for HA_PEO@PC.

For this sample, the value of scavenging ability of the DPPH radical was 58.35 ± 6%, and for the ABTS, 51.43 ± 9%. This property may be the result of *Calendula officinalis* infusion’s high flavonoid content, as the propolis containing samples presented significantly less activity against DPPH and ABTS (18.60 ± 8% for the HA_PEO@IP against DPPH and 28.79 ± 2% for HA_PEO@IP against ABTS). HA_PEO@ML presented the best results in the case of the PRAP assay, a fact also supported by literature data [51] and based on the phenolic compounds content.

#### 3.2.5. In Vitro Cytotoxicity Assay

In vitro biocompatibility of HA_PEO, HA_PEO@ML, HA_PEO@PC, and HA_PEO@IP was assessed by MTS assay after 72 h incubation with different extract concentrations (500, 250, 125, and 62.5 µg/mL) or fresh complete medium. The results showed that NFs extracts were not cytotoxic at concentrations up to 500 µg/mL (Figure 7) and are therefore in agreement with ISO 10993-5 guidelines [52]. Furthermore, there was no significant difference in cell viability between NFs extracts at 125 µg/mL. Additionally, HA_PEO@PC stimulates normal fibroblasts’ proliferation by 21% at concentrations of 250 µg/mL and by 37% at 500 µg/mL (Figure 7).

#### 3.2.6. Antimicrobial Assay

The antibacterial and antifungal evaluation of HA-PEO-NFs assayed the area of inhibition against Gram-negative (*Escherichia coli, Pseudomonas aeruginosa*), Gram-positive (*Staphylococcus aureus*), and *Candida albicans* strains. These strains are a frequent source of infection and may afterwards interfere with wound healing. A high percentage of clinically acquired infections are due to *Staphylococcus aureus*. *Escherichia coli* is the main cause of infection in burn wounds. Manuka honey was studied for its benefits in nanofibrous materials with the help of inhibition-zone determination [53], and its effect is largely attributed to its methylglyoxal constituent. Different activity against the bacterial strains used may be the result of the different cell wall structure of the Gram-positive and Gram-negative bacteria. The antimicrobial effect of HA-PEO-NFs expressed as diameters of the inhibition area (mm) are presented in Table 3. The samples displayed good antibacterial effects, especially against *Staphylococcus aureus*. The most active were HA_PEO@PC and HA_PEO@ML with their effect being half of Ciprofloxacin, used as control. Concerning the antifungal assessment, no sample displayed any effect. Considering the infection risk of wounds, the dressing materials which possess antibacterial activity are of high interest given their beneficial effects toward wound healing.

All designed HA-PEO-ACs NFs display promising results referring to in vitro cell toxicity, antimicrobial effect, and antioxidant potential. This can be explained as a cumulative effect of the polymeric matrix and the incorporated ACs. Taken separately, the ACs have proven their wound-healing effects [25,54,55,56]. The novelty of this work is based on combination of different ACs in order have a synergetic effect and so to improve the healing effects of HA. The results proved this hypothesis which so support the wound-dressing potential of HA-PEO-ACs NFs. The antimicrobial effects recorded for HA_PEO@PC and HA_PEO@ML against *Staphylococcus aureus* are even improved compared with other previously reported NFs. Our results are in agreement with other reported data for Manuka honey NFs based on cellulose acetate [53], polyvinylalcohol [57], and silk fibroin [22], which had a favorable effect for fibroblasts development but were detrimental to pathogenic bacteria proliferation. Except for the Manuka honey–polyvinylalcohol NFs which were tested for wound-healing effects in combination with pomegranate extract, no other reported Manuka honey NFs were tested as dressing materials. In addition, HA_PEO@PC and HA_PEO@ML were proved to be active also on *Pseudomonas aeruginosa* strain, while other research did not study the effect on this strain. This is important novelty element because *Pseudomonas aeruginosa* is a common strain causing nosocomial infection [58,59], which can aggravate and delay the wound healing. The presence of L-arginine together with Manuka honey into the polymer matrix further promotes the possibility of proline synthesis at the wound site and late collagen synthesis with the prospects of a speedier wound recovery [27]. Insulin is a signaling molecule with key role in cell migration, proliferation, and differentiation which translates into a beneficial effect on the fibroblasts [60]. Moreover, the association with propolis enhances insulin’s properties regarding proliferation, a content of less that 10% propolis being favorable in this matter [61], which was proved also by our results. A correlation between the active components of *Calendula officinalis* has been previously reported in the literature, based on its use in inflammatory cutaneous pathologies, burns, or sun exposure-caused erythema [31,32]. Referring to the composition of *Calendula officinalis* extract is reported that the essential oil has antibacterial and antifungal effects, the flavonoids present anti-inflammatory properties, and the saponins displayed anti-inflammatory, antibacterial, and antifungal properties, while the carotenoids are the components responsible for cell-interactive properties [62]. These constituents form a phyto-complex which acts synergic for its biological effects, proving antioxidant, cell-compatible, and antimicrobial effects.

## 4. Conclusions

The new HA-PEO-NFs, which have inglobated different ACs such as Manuka honey, propolis, *Calendula officinalis*, insulin, and L-arginine, have been prepared and characterized in terms of physicochemical and structural features, including morphology, fiber diameters, FTIR spectra, and WVTR. The thin NFs (ranging from 156 to 230 nm) exhibited suitable WVTR values (ranging from 4634 ± 4 g/m^2^ 24 h to 5122.6 ± 2 g/m^2^ 24 h) and could maintain a moist environment favorable for wound healing. That means that obtained NFs form vapor-permeable and microbe-preventive membranes. The biological evaluation, based on in vitroassays, proved that the antimicrobial and antioxidant effects of the developed NFs are due to the ACs, since HA_PEO matrix does not exhibit any effect. Moreover, the addition of the ACs maintained the cytocompatibility of the polymeric membranes. In this respect, the obtained NFs may be considered safe nanocarriers for natural active components. A future step in improving the characteristics of the obtained NFs will be to assure the support as proper wound dressings. Taking into consideration that chemical or physical crosslinking of porous materials improves stability in aqueous media, the crosslinking of HA-based NFs with EDC/NHS will be performed. Even more, layer-by-layer dressings can be formulated based on the results obtained. In addition, in vitro degradation assay, using phosphate buffered solution, pH = 7.4 (PBS), and pseudo extracellular fluid solution (PECF), will be performed. Based on in vitro degradation assay the molecular weight distribution of the HA in the corresponding experimental conditions will be also determined.

The results of our study suggest that the developed formulations had a synergistic effect on behalf of the ACs, especially for HA_PEO@PC, that promoted cellular proliferation. This formulation may be considered as an effective and safe candidate for wound healing, and the results encourage us to include it in anin vivo study using an acute wound model (surgical excision) induced to rats.

## Figures and Tables

**Figure 1 polymers-13-01291-f001:**
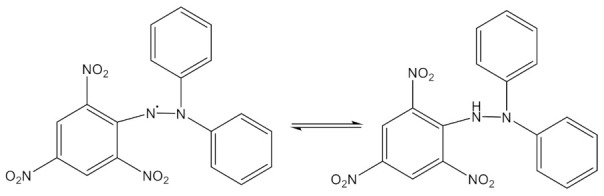
The reduction reaction of DPPH.

**Figure 2 polymers-13-01291-f002:**
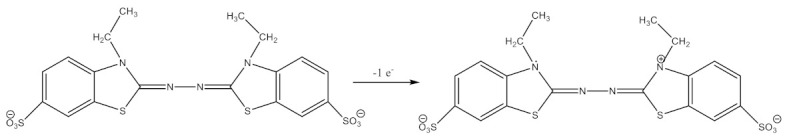
The generation reaction of ABTS^•+^.

**Figure 3 polymers-13-01291-f003:**
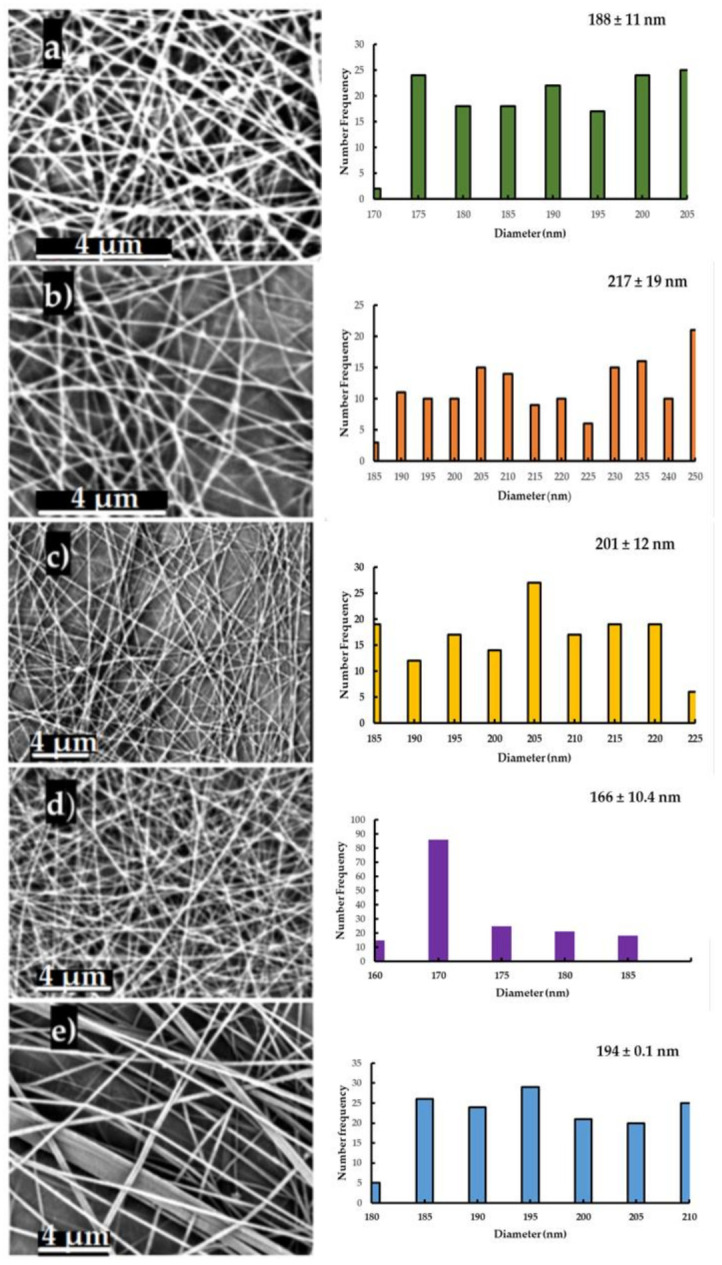
SEM micrographs, fiber distribution (number frequency) and average diameter (nm) of HA_PEO-NFs: HA_PEO@P (**a**), HA_PEO@IP (**b**), HA_PEO@PC (**c**), HA_PEO@ML (**d**), HA_PEO (**e**).

**Figure 4 polymers-13-01291-f004:**
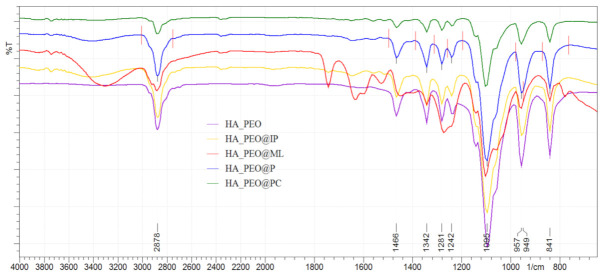
Fourier transform infrared spectroscopy (FTIR) assay of HA-PEO-ACs nanofibers (NFs).

**Figure 5 polymers-13-01291-f005:**
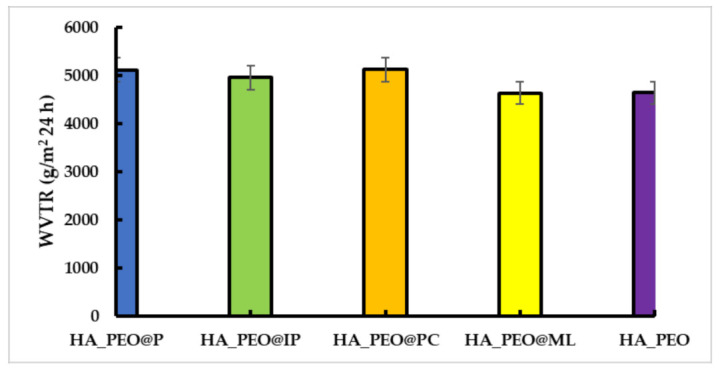
Water-vapor transmission rate (WVTR) assay of HA-PEO-NFs (expressed as g/m^2^ 24 h).

**Figure 6 polymers-13-01291-f006:**
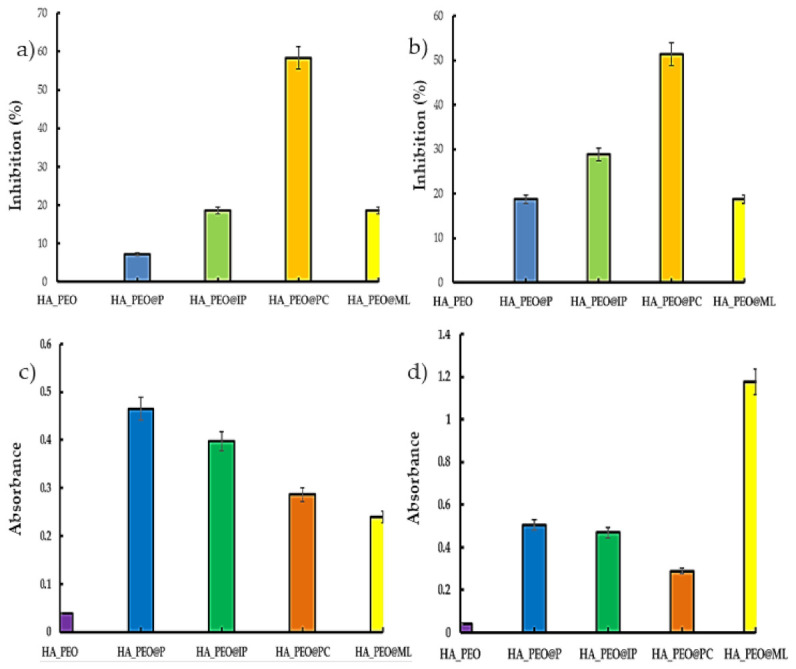
The antioxidant effects of HA-PEO-NFs: DPPH (**a**), ABTS (**b**), ferric-reducing antioxidant power (FRAP) (**c**), and phosphomolybdenum-reducing antioxidant power (PRAP) (**d**).

**Figure 7 polymers-13-01291-f007:**
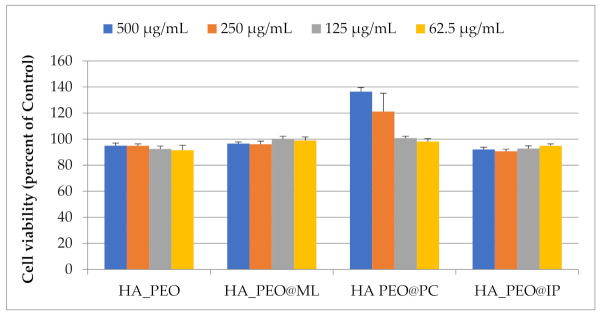
Cell viability (%) of HA-PEO-NFs, at different concentrations tested on normal human dermal fibroblasts.

**Table 1 polymers-13-01291-t001:** Parameters of hyaluronic acid-polyethylene oxide-active components (HA-PEO-ACs) blending electrospinning solutions.

HA-PEO-NFs	Solvent	ACs
HA_PEO@P	normal saline	P: 7% (wt/*V*) ^1^
HA_PEO@IP	normal saline	I: 100 IU/mL P: 7% (wt/*V*)
HA_PEO@PC	*Calendula officinalis* infusion	P: 7% (wt/*V*)
HA_PEO@ML	normal saline	M: 7% (wt/*V*) ^1^ L: 7%(wt/*V*) ^1^

^1^ Higher concentrations led to dripping and bead formation.

**Table 2 polymers-13-01291-t002:** Viscosity parameters of HA-PEO-ACs blending electrospinning solutions.

HA-PEO-ACs	Ostwald I Model		Carreau–Yassuda Model Zero-Shear Viscosity, η_0_ (Pa∙s)	Apparent Viscosity, at 100 s^−1^ (η_a,100_, Pa∙s)
	Flow Behavior Index, a (K)	Consistency Coefficient, b (n)		
HA_PEO@IP	2.5	0.56	1.5	0.4
HA_PEO@ML	3.8	0.59	1.7	1.5
HA_PEO@P	2.3	0.57	1.2	0.4
HA_PEO@PC	1.9	0.58	0.9	0.3

**Table 3 polymers-13-01291-t003:** Antimicrobial effects of the HA-PEO-NFs tested on bacterial and fungal strains.

Sample	Diameter of Inhibition Area (mm)			
	*S. aureus* ATCC 25923	*E. coli* ATCC 25922	*P. aeruginosa* ATCC 27853	*C. albicans* ATCC 10231
HA_PEO@P	11	0	0	0
HA_PEO@IP	11	0	0	0
HA_PEO@PC	17	16	14	0
HA_PEO@ML	12	15	10	0
HA_PEO	0	0	0	0
CIP (5 µg/disc)	26	28	30	*Nt*
VRC (1 µg/disc)	*nt*	*nt*	*nt*	27

## Data Availability

The data presented in this study are available on request from the corresponding author.

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
