# Peer review of "New Hyaluronic Acid/Polyethylene Oxide-Based Electrospun Nanofibers: Design, Characterization and In Vitro Biological Evaluation"

_polymers, 2021, doi:10.3390/polym13081291_

Round 1

Reviewer 1 Report

Specific comments:

  1. Across the text (lines 27, 287-292, 320-323, 333-337, 375, Figure 1): The standard deviation should be expressed as ONE significant figure; that is, unless the number is between 11 and 19 times some power of ten, in which case you can use two significant figures. The mean value should be rounded off at the decimal place corresponding to the last significant digit of its standard deviation. E.g., 217±18.98 (line 27) should be presented as 217±19; 4636.33 ± 4.2 (lines 322-323) should be presented as 4636 ± 4.
  2. Line 62: Remove “non-sulfated”. Нyaluronic acid is, in fact, a non-sulfated polysaccharide, but it is not clear to me why this should be emphasized.
  3. Lines 72 and 76: Consider citing here the recently published paper on this topic (Petrova et al. Electrospun bilayer chitosan/hyaluronan material and its compatibility with mesenchymal stem cells. Materials, 2019, 12(12), 2016, DOI: 3390/ma12122016).
  4. Line 115: The properties of hyaluronan are very dependent on its molecular weight (including viscosity, surface tension, and electrospinnability); therefore, it must be carefully characterized regarding its molecular weight by viscometry, light scattering, or size exclusion chromatography.
  5. Line 142: What is the authors' motivation for using normal saline to prepare electrospinning solutions? Why is water not suitable for this purpose?
  6. Lines 142, 364: What does the acronym CS (CS and CS_PEO@PC) stand for?
  7. Figure 1: Quality of Figure 1 must be improved; the scale bars are hardly visible.
  8. “3.2.2. Solid State Properties” The name of the sub-section does not correspond to its content. It should be renamed.
  9. Figure 2: It would be useful to add the HA_PEO spectrum for comparison.
  10. Section 3.2.2. Solid State Properties: The description and discussion of the FTIR spectra (lines 298-311) are very confusing and must be completely revised. The absorption band at 2885 cm−1 cannot be associated with -OH stretching (this might be a C-H stretching which falls at around 3000-2840 for alkanes), as well as 1350-1400 cm-1 with the carboxyl group (for the carboxylATE group there should be two asymmetric and symmetric stretches at around 1650-1540 and 1450-1360, respectively). Are amide I and II bands observed for hyaluronan? What do you mean by "asymmetric bending of CH" given that the CH group has no bending vibrations. This whole section should be carefully revised!
  11. Line 374: No one views the FTIR as a “solid state properties”. Please, revise.

Author Response

Dear Reviewer,

The authors thank to Reviewer for evaluation of the quality of our manuscript and for his constructive suggestions and interesting comments. All changes in the manuscript were highlighted using track changes and  for the detailed responses please see the attached document.

Thank you,

Sincerely yours,

Prof. Dr. Lenuta Profire

University of Medicine and Pharmacy „Grigore T. Popa” of Iasi

16 University Street

700115, Iasi, Romania

e-mail:  lenuta.profire@umfiasi.ro

Reviewer 2 Report

The dynamic process of wound healing needs the support of such a dressing that can mimic and promote the natural process of healing. While it is important to keep a moist environment that can aid healing, the designing of a dressing that would meet the demanding qualities for maintaining cell growth whilst ensuring their differentiation, represents still an important aspect of biomaterial engineering. Nanofibersare relatively new porous systems currently investigated  for their multiple purposes. In this paper, different natural active components such as propolis, manuka honey, insulin, L-arginine and Calendula officinalis infusion were included into hyaluronic acid/poly(ethylene)oxide based electrospun nanofiber membranes to design innovative wound dressing biomaterials. Based on their unique properties (large surface and high porosity), nanofibers act as excellent extracellular matrices which enhance the tissue formation. Chemical composition was proved by Fourier Transform Infrared Spectroscopy, which indicated successful incorporation of the active components. The topic is important, the results are interesting and the methodology followed is appropriate, while the content falls well within the scope of this Journal. In general the paper makes fair impression and my recommendation is that it merits publication in this Journal, after the following major revision:

  1. The authors need to reorganize the current introduction, which normally consists of three parts at least: background, literature review, brief of the proposed work. The current one is nothing but a literature review. Why their work is important comparing to previous reports? I think this is essential to keep the interest of the reader.
  2. Materials and Methods part. Although the results look “making sense”, the current form reads like a simple lab report. The authors should dig deeper in the results by presenting some in-depth discussion.
  3. In Fig. 2, the authors should give the explanations for the difference of data collected from different sources.
  4. The nanofiber membranes with propolis and Calendula officinalis showed best antioxidant activity, cytocompatibility and antimicrobial properties against pathogen strains Staphylococcus aureus, Escherichia coli and Pseudomonas aeruginosa, and had an average diameter of 217±18.98 nm with smooth surface aspect. The authors should give some explanation on above conclusions and data.
  5. In this paper, surface morphology and fiber diameter were analyzed by scanning electron microscopy. Fiber morphology was carried out using scanning electron microscopy. Fiber diameter and distribution were analyzed using the phenom fiber metric software. The present work mainly focuses on lab work. It does not necessarily imply that the theoretic work (modeling) is not important. The authors omit this part during the current literature review, which should include a brief review of the theoretic work after revision. In the theoretic perspective, fractal theory is a very important tool, which can be used to investigate the surface morphology and fiber diameter of fibers,  (see [A fractal model for capillary flow through a single tortuous capillary with roughened surfaces in fibrous porous media, Fractals, 2021, 29(1):2150017; Fractals, 2019, 27(7): 1950116]). Authors should introduce some related knowledge to readers. I think this is essential to keep the interest of the reader.
  6. Please, expand the conclusions in relation to the specific goals and the future work.

Author Response

Dear Reviewer,

The authors thank to Reviewer for evaluation of the quality of our manuscript and for his constructive suggestions and interesting comments. All changes in the manuscript were highlighted using track changes and for detailed responses please see the attached document.

Thank you!

Sincerely yours,

Prof. Dr. Lenuta Profire

University of Medicine and Pharmacy „Grigore T. Popa” of Iasi

16 University Street

700115, Iasi, Romania

e-mail:  lenuta.profire@umfiasi.ro

Reviewer 3 Report

This work presents a fascinating trend of a biomedical group writing about nanofibers' application as a wound dressing. Instead of laborious work not connected to the medical background, we have a piece of work from the people who know how to use the materials.
Unfortunately, the group contained no chemists or electrospinning specialists, so some additions should be material should be added before the publication in the "Polymers". First of all - many problems were caused by using "salt solution from local pharmacy". Such a statement could actually disqualify the work from the beginning. Please change this. Some conclusions are wrong because using salt solutions in electrospinning usually causes problems. Also, using hyaluronate fibers without crosslinker will cause the micro-and nanofibers to dissolve instead of serving as a wound dressing - please add a future work section with the crosslinking of the mats (typically by EDC or NHS). There is a lot of analytical chemistry work done with no mentioning of chemistry. Please add schemes of chemical reactions at all sections mentioning that. Please add proper figures description - what compound was used - instead of samples-like coding (e.g. Propolis NF instead of HA_PEO@P) to facilitate reading. The work is fascinating, and I do like to encourage authors to continue. And to add some chemist and electrospinning specialists to the group. 

Author Response

Dear Reviewer,

The authors thank to Reviewer for evaluation of the quality of our manuscript and for his constructive suggestions and interesting comments. All changes in the manuscript were highlighted using track changes and for the detailed responses please see the attached document.

Thank you!

Sincerely yours,

Prof. Dr. Lenuta Profire

University of Medicine and Pharmacy „Grigore T. Popa” of Iasi

16 University Street

700115, Iasi, Romania

e-mail:  lenuta.profire@umfiasi.ro

Round 2

Reviewer 1 Report

The authors have successfully addressed most of my concerns, improving the manuscript with their edits. However, I want to draw the authors' attention to the importance of characterizing all the examined polymers by molecular weight and molecular weight distribution. Unfortunately, the authors still do not present the molecular weight of hyaluronan.

Author Response

Dear Sir/Madame

Thank you very much for your valuable recommendations. The requested data were included in the manuscript. and in the attached document.

Sincerely yours,

Prof. Dr. Lenuta Profire

University of Medicine and Pharmacy „Grigore T. Popa” of Iasi

16 University Street

700115, Iasi, Romania

e-mail:  lenuta.profire@umfiasi.ro

Reviewer 2 Report

It is ok.

Author Response

Dear Sir/Madame

Thank you very much for your accepting.

Sincerely yours,

Prof. Dr. Lenuta Profire

University of Medicine and Pharmacy „Grigore T. Popa” of Iasi

16 University Street

700115, Iasi, Romania

e-mail:  lenuta.profire@umfiasi.ro